# Record high-$T_c$ and large practical utilization level of electric polarization in metal-free molecular antiferroelectric solid solutions

Haojie Xu[1,2], Wuqian Guo[1,2], Yu Ma[1,2], Yi Liu[1,2], Xinxin Hu[1,2], Lina Hua[1], Shiguo Han[1,3], Xitao Liu [1,3], Junhua Luo [1,2,3] ✉ & Zhihua Sun [1,2,3] ✉

Metal-free antiferroelectric materials are holding a promise for energy storage application, owing to their unique merits of wearability, environmental friendliness, and structure tunability. Despite receiving great interests, metal-free antiferroelectrics are quite limited and it is a challenge to acquire new soft antiferroelectric candidates. Here, we have successfully exploited binary $CMBr_xI_{1-x}$ and $CMBr_xCl_{1-x}$ solid solution as single crystals ($0 \leq x \leq 1$, where CM is cyclohexylmethylammonium). A molecule-level modification can effectively enhance Curie temperature. Emphatically, the binary CM-chloride salt shows the highest antiferroelectric-to-paraelectric Curie temperature of ~453 K among the known molecular antiferroelectrics. Its characteristic double electrical hysteresis loops provide a large electric polarization up to ~11.4 μC/cm², which endows notable energy storage behaviors. To our best knowledge, this work provides an effective solid-solution methodology to the targeted design of new metal-free antiferroelectric candidates toward biocompatible energy storage devices.

Electroactive materials that enable to store and release electrical energy have been widely used as basic ingredients of energy-storage systems, electromagnetic devices, and high-power microwaves[1-4]. Among them, antiferroelectric (AFE) materials, featured by the antipolar array of electric dipoles inside the neighboring sublattices[5,6], have long been a fascinating branch for energy storage. Particularly, such initially antiparallel dipoles can be switched into parallel alignment under high electric fields, namely, AFE-to-ferroelectric (FE) structural transformation[7,8]. Compared with linear dielectrics and ferroelectrics, this distinctive electric-induced phase transition in AFEs leads to characteristic double P–E hysteresis loops, thereby allowing to store a larger amount of electric energy[9]. The current mainstream of AFE materials for energy storage is still dominated by inorganic oxides containing heavy metals[10,11], such as $PbZrO_3$, (Ba, Na)$TiO_3$, and $AgNbO_3$, of which antiferroelectricity stems from ionic displacement and their

optimized performances highly depend on chemical doping to these archetypes. To work toward a sustainable and green society, nevertheless, metal-free molecular AFEs are booming as an alternative family. For example, purely organic squaric acid (SQA) shows a comparable storage efficiency with inorganic oxides[12], and a giant AFE-related electrostriction was observed in 2-trifluoromethylnaphthimidazole[13]. However, the majority of existing molecular AFEs suffer from a quite low Curie temperature ($T_c$), such as $NH_4H_2PO_4$ ($T_c = 147$ K)[14], 5,5′-dimethyl-2,2′-bipyridinium chloranilate ($T_c = 318$ K), and SQA ($T_c = 373$ K)[7], indicating the low operating temperature condition (<100 °C) that is not suitable for practical applications[15-17]. Therefore, it is of significance and urgent to exploit new metal-free molecular AFE candidates with high $T_c$ and large spontaneous polarization ($P_s$).

Solid solution methodology has been substantially devoted to optimizing and modifying the physical properties of ferroic-ordered

[1]State Key Laboratory of Structural Chemistry, Fujian Institute of Research on the Structure of Matter, Chinese Academy of Sciences, Fuzhou, Fujian 350002, People's Republic of China. [2]University of Chinese Academy of Sciences, Chinese Academy of Sciences, 100039 Beijing, People's Republic of China. [3]Fujian Science & Technology Innovation Laboratory for Optoelectronic Information of China, 350108 Fuzhou, Fujian, People's Republic of China. ✉e-mail: jhluo@fjirsm.ac.cn; sunzhihua@fjirsm.ac.cn

materials, including both inorganic oxides and inorganic-organic hybrid ferroelectrics[18–20]. For instance, AgNbO₃ crystals exhibit quite weak room-temperature ferroelectricity, while the metal-doping of La³⁺ and Sm³⁺ results in strong AFE orders in AgNbO₃-based ceramics that facilitate high-efficiency energy storage. Recently, solid solutions of inorganic-organic hybrid ferroelectrics, $(TMFM)_x(TMCM)_{1-x}CdCl_3$ $(0 \leq x \leq 1)$, were successfully achieved through cation modifications, which display a huge enhancement of piezoelectric coefficients up to 1540 pC/N near the morphotropic phase boundaries. For purely metal-free systems, dynamic orderings of molecular moieties usually dominate the potential energy barriers that are crucial to modulating $T_c$ for phase transitions[21–23]. Nevertheless, the formation of metal-free solid solutions is quite challenging, since the mixing of different molecules in a single homogeneous phase closely involve with their spatial orientation and configurational distortion. This probably accounts for the lacking of high-$T_c$ molecular AFEs achieved by the solid solution method. In this context, we here attempt to construct new metal-free AFEs through the solid solution methodology, of which the countered anions serve as variables to form the multi-composition solid solutions.

Here, we have successfully obtained new metal-free molecular AFEs in a series of binary $CMBr_xI_{1-x}$ and $CMBr_xCl_{1-x}$ solid solutions $(0 \leq x \leq 1$, where CM is cyclohexylmethylammonium) as single-phase homogeneous crystals. A subtle modification of halogen-bonded dimers raises phase transition energy barriers that result in the enhanced $T_c$. Among them, the CM-chloride crystal (CMC, $x = 0$ of $CMBr_xCl_{1-x}$) exhibits an AFE-to-paraelectric (PE) phase transition at ~453 K, falling in the range of the record-high values among molecular AFEs. Particularly, a large practical utilization level of $P_s$ ~ 11.4 μC/cm² in CMC, as deduced from the double $P$–$E$ hysteresis loops, reveals its potentials for energy storage application. This is the first time to achieve such a record high-$T_c$ and large practical utilization level of $P_s$ in the molecular AFE systems. This result sheds light on exploiting metal-free AFE candidates for future biocompatible energy storage.

## Results and discussion

### Molecule-level design of metal-free AFEs

The family of $CMBr_xI_{1-x}$ and $CMBr_xCl_{1-x}$ solid solutions adopt a hydrogen-bonding dimer motif, resembling the binary stator–rotator system in AFE-to-PE phase transition (Fig. 1a). Organic CM⁺ cation is the dynamically active moiety with order–disorder genius like "rotator", while halogen anions linked to cations via N·H·X hydrogen bonds stabilize the structure. The cooperativity of dipolar components in the adjacent crystal lattices afford the power source to create long-range AFE orders, mainly determined by the cationic reorientation. A hint is the stator–rotator interactions play a crucial role to increase energy barriers, as well as the realization of high-$T_c$ AFE orders. Notably, this binary solid-solution family has a delicate compositional dependence on intermolecular H-bonds, as shown by Hirshfeld surface, which results in the enhanced $T_c$. Taking CMB as a prototype[24], the height of energy barriers can be successfully regulated by I/Br/Cl halogen modification (Fig. 1b, c), which leads to a wide tunable rang of $T_c$ from 324 K of CMI to 453 K of CMC.

### Temperature-dependent crystal structure transition

Colorless crystals of $CMBr_xI_{1-x}$ and $CMBr_xCl_{1-x}$ $(0 \leq x \leq 1)$ were synthesized from aqueous solutions with the stoichiometric ratios (Supplementary Fig. 1). Results of X-ray photoelectron spectroscopy and Powder X-ray diffraction of Cl/Br/I elements coincide with the theoretical calculations (Supplementary Figs. 2, 3). Structure analyses reveal that both CMC and CMI, isostructural to our reported CMB analog, crystallize in orthogonal system with the point group *mmm* at room temperature, thus enabling the compatibility of solid solutions. Figure 2a shows the basic unit consisting of H-bonding dimers, resembling the famous stator–rotator systems[25–29], in which the orientations of H-bonding dimers are arranged in the opposite directions (Fig. 2b), which eliminate bulk polarization of unit cell and demonstrate possible AFE order. Moreover, the N-H···X distances between stator and rotator are dependent on the halogen sizes. The N···Cl distance of CMC ≈ 3.148 Å is much shorter than N···Br ≈ 3.300 Å and N···I ≈ 3.529 Å. Such different interactions related to ion radius and electronegativity cause a significant change of $T_c$. Take CMC as an example, CM⁺ cations become highly disordered at high-temperature paraelectric phase accompanied by a halving of the volume (Fig. 2c–f). Dynamic ordering and reorientation of C-N bonds and six-member rings of two adjacent bipartite arrays require much more energy to overcome barriers, corresponding to the higher $T_c$.

The $CMBr_xI_{1-x}$ and $CMBr_xCl_{1-x}$ solid solutions have an isostructural high-temperature phase (HTP) with the nonpolar space group P4/*nmm* (point group 4/*mmm*), in which organic CM⁺ cations

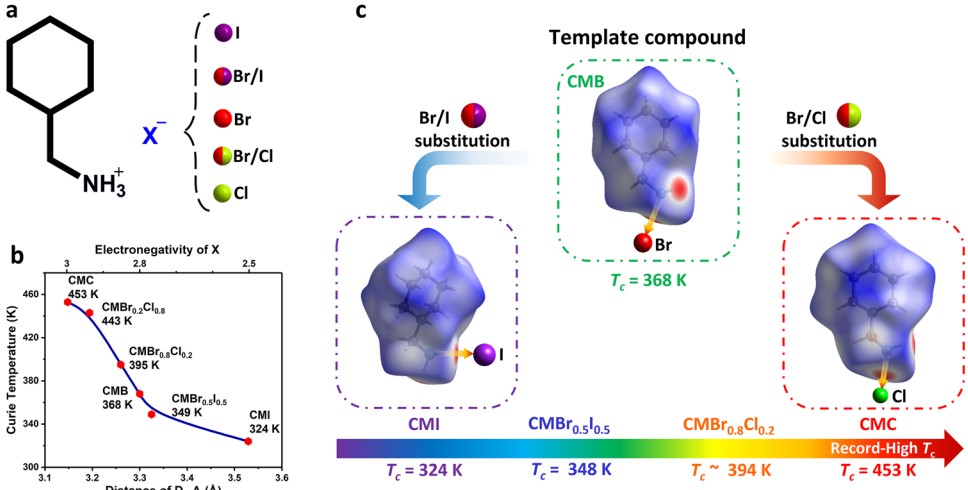

**Fig. 1 | Diagram for the molecule-level design of metal-free molecular AFEs in a series of CMBrxI₁₋ₓ and CMBrxCl₁₋ₓ (0 ≤ x ≤ 1) solid solutions. a** Basic structures for solid solutions. **b** The relationship between Curie temperatures and D···A distances of hydrogen bonds, where donor (D) is N of organic CM⁺ cation and acceptor (A) is halogen (X). The D···A distances can be obtained from the crystal structures; the averaged electronegativity of the solid solution is estimated from the ratio x based on the known Cl/Br/I. **c** $T_c$ for the solid solutions is dramatically enhanced through the gradual substitution, the background refers to Hirshfeld surface to quantify intermolecular interactions.

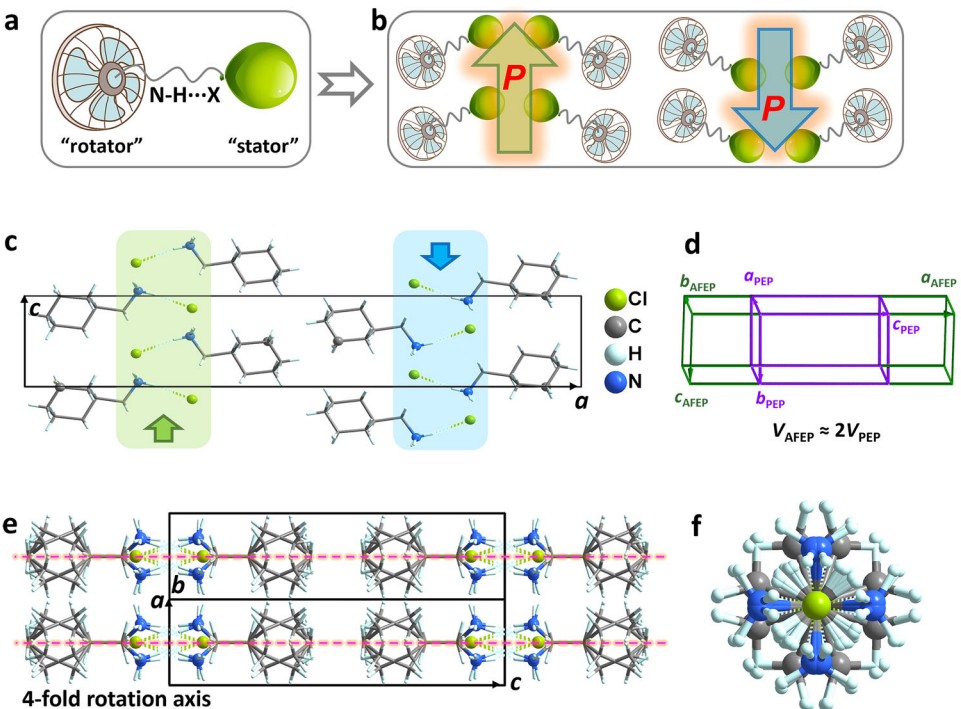

**Fig. 2 | Crystal structures of binary solid solutions. a, b** Schematic diagram of CMBr$_x$I$_{1-x}$ and CMBr$_x$Cl$_{1-x}$ binary solid solutions. The organic cations afford a driving force to trigger phase transitions. **c** Projection of AFE structure of CMC at LTP. **d** The cell edges of both the AFEP and PEP were drawn to show volume multiplication of AFE. **e, f** Structure packing of CMC at the paraelectric HTP *P*4/ *nmm*, showing the complete disordering of organic CM cations.

are rotationally disordered and adopt 4-fold rotation axis ($C_4$) symmetry (Supplementary Figs. 4, 5). The C-N bonds of cations occupy eight equivalent sites that obey the basic requirement of crystallographic symmetry. Structure comparison reveals that the order–disordering accounts for their phase transitions and the vanishing of bulk polarization. The halogen anions of these solid solutions exhibit relative atomic displacements in FEP, thus coinciding with the variation of crystal symmetry[24]. Hence, the origin of AFE-FE-PE phase transitions can ascribe to the collaboration between organic cationic ordering and anionic displacement.

**Phase transitions and related properties**

To verify the occurrence of structural phase transitions, we analyzed differential scanning calorimetry (DSC) on the typical compounds in the CMBr$_x$I$_{1-x}$ and CMBr$_x$Cl$_{1-x}$ solid solutions (Fig. 3a and Supplementary Fig. 6). Sequences of sharp endo/exothermic peaks in the heating/cooling modes indicate that their $T_c$ values can be efficiently regulated between 324 K (CMI) and 453 K (CMC), which depend on the compositions of anionic halogen. It is intriguing to note that the CMB and CMBr$_{0.8}$Cl$_{0.2}$ solid solutions undergo two phase transitions, of which the intermediate phase is determined as a polar space group of *P*4*mm*. Furthermore, strong λ-peak anomalies are clearly observed in the temperature-dependent dielectric measurement (Fig. 3b). With the composition changing from I to Br to Cl, the $T_c$ for observing sharp dielectric anomalies is also increasing, in agreement with DSC and structure analyses (Fig. 3c). Especially, the monocomposition CMC exhibits both the highest $T_c$ and strongest dielectric response, including low dielectric loss (<0.01, at room temperature) and large dielectric constant ($\varepsilon'$) variation (Fig. 3d). This should be reminiscent of its significant AFE order. Subsequently, we further determine the phase structures as a function of temperature and the halogen composition. As depicted in Fig. 3e and Supplementary Fig. 7, CMC has the record-high $T_c$ among the metal-free molecular AFEs at 453 K, e.g. 5,5'-dimethyl-2,2'-bipyridinium chloranilate (~318 K), SQA (~373 K), and catches up with some oxide counterparts, such

as Pb$_{0.97}$La$_{0.02}$(Zr$_{0.50}$Sn$_{0.43}$Ti$_{0.07}$)O$_3$ (~433 K) and Pb$_{0.93}$Ba$_{0.04}$-La$_{0.02}$Zr$_{0.95}$Ti$_{0.05}$O$_3$ (~ 450 K)[12,30,31]. As far as we know, this is the first time to report metal-free molecular solid solutions with such a widely adjustable $T_c$, and CMC has the record-high $T_c$ which is well below the melting point (Fig. 3f), which probably knocks the door toward practical application.

**Antiferroelectric and energy storage behaviors**

Generally, the typical *P*–*E* hysteresis loop is the most convincing evidence to reveal bulk AFE and FE orders, which are microscopically manifested as dipoles flipping under external electric field. Therefore, we prepared electrodes by observing the optical axis and ferroelastic domains under a polarized light microscope to observe the characteristic *P*–*E* hysteresis loops from the current density versus electric field (*J*–*E*) curves for CMBr$_x$I$_{1-x}$ and CMBr$_x$Cl$_{1-x}$ solid solutions (Fig. 4a–c and Supplementary Fig. 9). Figure 4d depicts the selected members show obvious current peaks in the *J*–*E* curves at different temperatures. To emphasize, an additional current peak emerges in CMBr$_{0.2}$Cl$_{0.8}$, suggesting the coexistence of both AFE and FE orders that resemble many other inorganic solid solutions[32,33]. For CMC, two pairs of current peaks can also be clearly observed just below $T_c$, which act as the direct proof of AFE properties. From the hysteresis loops in Fig. 4e, the AFE $P_s$ of CMC is estimated as 11.4 μC/cm$^2$, much higher than other molecular AFEs, such as 2-trifluoromethylbenzimidazole (~5.6 μC/cm$^2$), (5,5'-dimethyl-2,2'-bipyridine) (chloranilic acid) (~4.9 μC/cm$^2$), and hybrid perovskite (isopentylammonium)$_2$CsPb$_2$Br$_7$ (~6 μC/cm$^2$)[12,34,35]. Besides, the coercive electric field ($E_c$) that defines the strength to reverse dipolar moments displays temperature-dependent behavior; for instance, the forward switching field ($E_A$, AFE-to-FEP transition) increases from 41 kV/cm at 443 K to 55 kV/cm at 433 K (Supplementary Fig. 8), and even higher $E_c$ is require to switch electric dipoles. Such large practical utilization level of $P_s$ and high $E_c$ provide great potential for CMC toward energy storage applications.

Above studies reveal that CMC is a metal-free molecular AFE with the record high-$T_c$ and large practical utilization level of $P_s$, which are

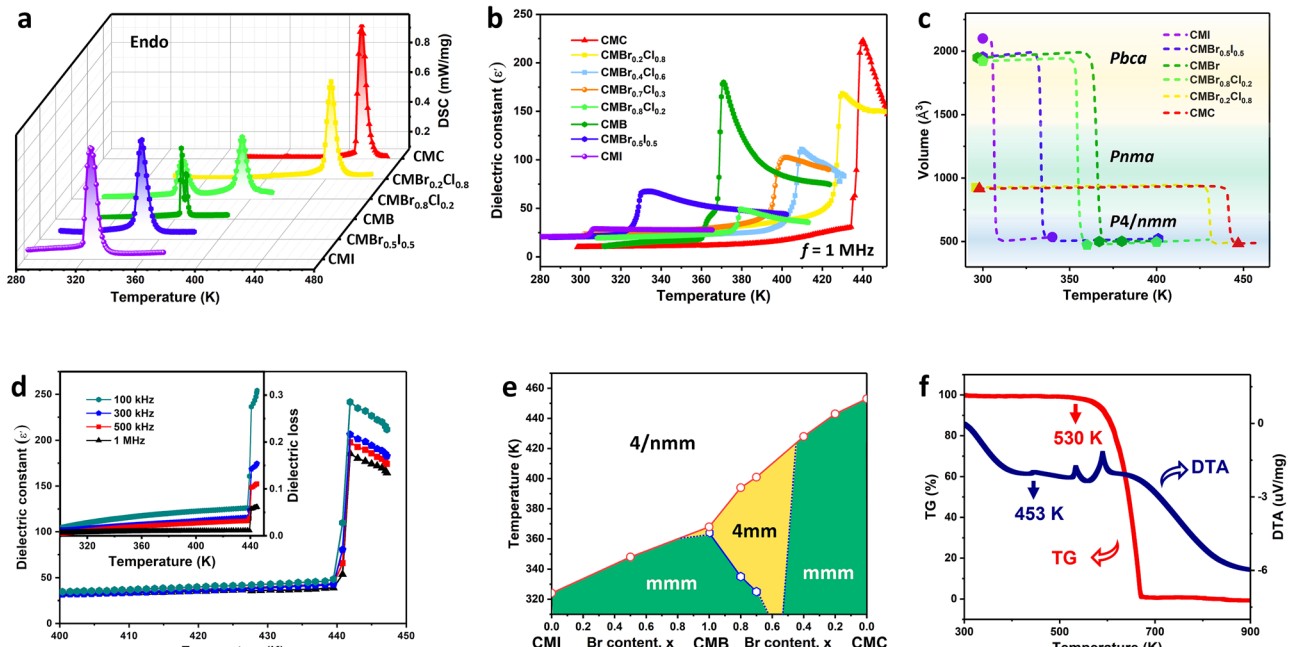

**Fig. 3 | CM content-induced phase transition in solid solutions. a** DSC curves of $CMBr_xI_{1-x}$ and $CMBr_xCl_{1-x}$ solid solutions. **b** Temperature variation of $\varepsilon'$ upon cooling. **c** Schematic diagram of temperature dependence versus cell lattice parameters obtained directly from the X-ray single-crystal diffraction, as depicted by the short dash lines. **d** Temperature dependence of $\varepsilon'$ and dielectric loss measured at different frequencies for CMC. **e** Phase diagram and the point groups of $CMBr_xI_{1-x}$ and $CMBr_xCl_{1-x}$ solid solutions. The dotted boundary is roughly estimated due to the difficulty of growing homogeneous crystals. **f** TG/DTA curves of CMC with heating rate 15 °C/min.

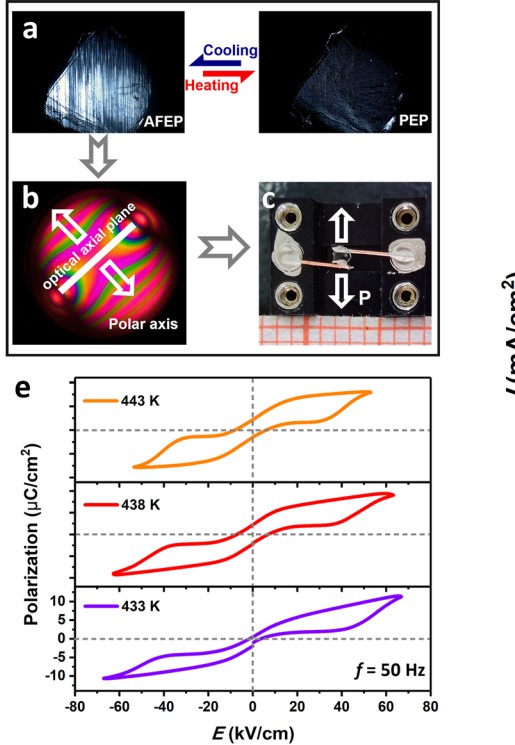

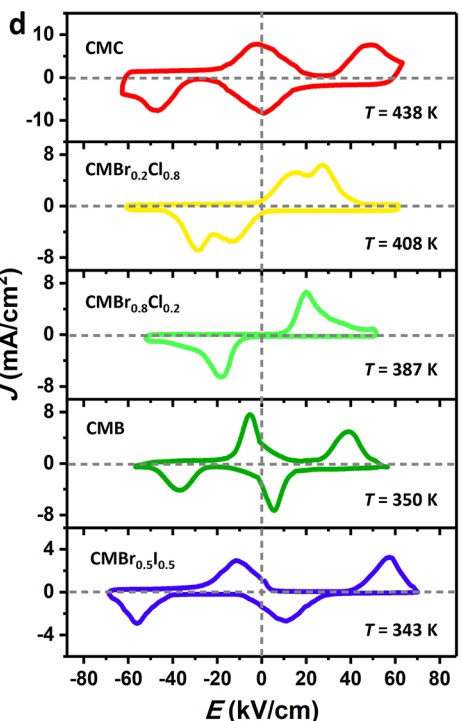

**Fig. 4 | AFE properties of $CMBr_xI_{1-x}$ and $CMBr_xCl_{1-x}$ solid solutions. a** Variable-temperature domains of **CMC**. **b**, **c** Preparation of electrodes according to the relationship between optical axis and the direction of electric polarization. **d** $J$–$E$ curves collected at different temperatures. **e** Variable-temperature double $P$–$E$ hysteresis loops of CMC.

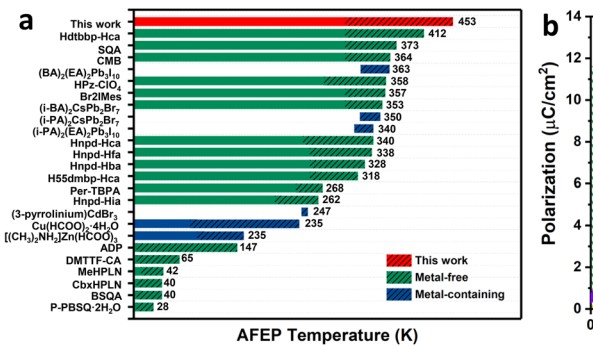
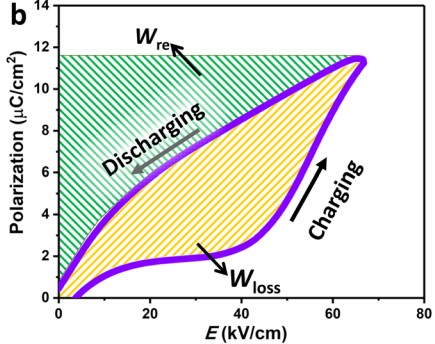

**Fig. 5 | AFE and energy storage related characteristics. a** AFEP Curie temperature ($T_c$) of CMC compared with some reported molecule-based AFE compounds. The columnar with twill background present the temperature range for experimentally measuring the antiferroelectric properties (for details, see Table S9). **b** Schematic illustration of the calculation of $W_{re}$ (green) and $W_{loss}$ (yellow) from the double $P$–$E$ hysteresis loop.

the essential criteria for energy storage application. Figure 5a presents the temperature range for retaining antiferroelectricity in several molecule-based AFE materials, of which CMC has a quite wide working temperature range. Furthermore, energy storage characters of CMC, including total energy density ($W_{st}$), recoverable energy density ($W_{re}$) and storage efficiency (η), are obtained from the integration of $P$–$E$ curves (Fig. 5b). The calculated $W_{re}$ and $W_{st}$ are 0.28 and 0.54 J/cm³, respectively; the corresponding η reaches up to 52% at 433 K, which is beyond that of $CMBr_{0.8}Cl_{0.2}$ (~20%, at 387 K). This efficiency is also comparable to some AFE oxides, e.g. $AgNbO_3$ (~40%), $Hf_{0.3}Zr_{0.7}O_2$ (~51%), $Pb_{0.97}La_{0.02}(Zr_{0.50}Sn_{0.39}Ti_{0.11})O_3$ (~50%), 0.95NBT-0.05BH (~48.2%), etc[17,36–38]. Given structural tunability and environment-friendly merits, these metal-free AFE solid solutions hold great application potentials in flexible and wearable devices.

To summarize, we have presented a family of metal-free molecular AFEs in $CMBr_xI_{1-x}$ and $CMBr_xCl_{1-x}$ ($0 \le x \le 1$) solid solutions by a precise molecule-level modification of halogen compositions. The CM-chloride salt displays the highest $T_c$ of molecular AFEs (~453 K) and large practical utilization level of $P_s$ up to 11.4 μC/cm². Such AFE attributes are unprecedented for the molecular AFEs, which allow notable energy storage performances in a relatively broad temperature range. This work provides guidance on the design of new metal-free AFE candidates, and might knock the door toward biocompatible, environmentally-friendly and flexible device applications.

## Methods

### Synthesis of CMX (X = Cl, Br, I)

The series of binary $CMBr_xI_{1-x}$ and $CMBr_xCl_{1-x}$ ($0 \le x \le 1$) solid solutions as single crystals were obtained via the temperature-cooling method. Cyclohexanemethylamine (1.5 ml, Aladdin, 99.9%) was dissolved in HX solution (20 ml, Aladdin, 48 wt% in water) by heating under strong stirring to colorless solution. Subsequently, the colorless sheet crystals were obtained by the temperature-cooling technology with a speed of -1 K/day and dried for subsequent use.

### Synthesis of $CMBr_xCl_{1-x}$ and $CMBr_xI_{1-x}$ ($0 < x < 1$)

Stoichiometric CMB ($x \times 5$ mmol) and CMC/CMI (($1-x$) × 5 mmol) were added into gamma-Butyrolactone/water (10 ml) to get a clear solution. After slow cooling at a rate of 1 K/day, the colorless sheet crystals can be obtained. The distribution of C, N, Cl, Br and I in the whole selected area reveals that the elements are uniformly dispersed on the single crystals. (Supplementary Figs. 10–12).

### Single crystal structure determination

Single crystal X-ray diffraction data of $CMBr_xCl_{1-x}$ and $CMBr_xI_{1-x}$ were collected on a Bruker D8 using the Mo Kα radiation at different temperatures. The direct method solved all the crystal structures and then refined them by the full-matrix least-squares refinements on $F^2$ using SHELXLTL software package. The non-hydrogen atoms were refined anisotropically based on all reflections with $I > 2\sigma (I)$.

### Thermal analysis

Differential scanning calorimetry (DSC) measurements were carried out using the NETZSCH DSC 200 F3 with a heating and cooling rate of 20 K/min under the nitrogen atmosphere. Thermogravimetric analysis (TGA) was measured using a STA449C Thermal Analyser ranging from room temperature to 900 °C with a heating rate of 15 °C/min.

### Dielectric and ferroelectric measurements

The dielectric constants ($\varepsilon'$) and dielectric loss were measured using the two-probe AC impedance method with an Impedance Analyzer (TH2828A). Moreover, the variable temperature current density versus electric field ($J$–$E$) curves and the polarization versus electric field ($P$–$E$) hysteresis loops were recorded with a ferroelectric analyzer (Radiant Precision Premier II).

### Calculation of energy storage properties

Total energy density $W_{st}$, recoverable energy density $W_{re}$ and energy density efficiency η can be estimated by the following equations:

$$W_{st} = \int_0^{P_{max}} E dP \text{(upon charging)} \tag{1}$$

$$W_{re} = - \int_{P_{max}}^{P_r} E dP \text{(upon discharging)} \tag{2}$$

η = $W_{st}/W_{re}$, and $W_{loss} = W_{st}-W_{re}$, where $E$ is the applied electric field, $P_{max}$ and $P_r$ represent the maximum and remnant polarization, respectively. These energy storage characteristics can be obtained from the integration of $P$–$E$ curves.

### Reporting summary

Further information on research design is available in the Nature Research Reporting Summary linked to this article.

## Data availability

All relevant data are presented via this publication and Supplementary Information. The X-ray crystallographic coordinates for structures reported in this study have been deposited at the Cambridge Crystallographic Data Centre (CCDC), under deposition numbers: CCDC 2167457-2167467. These data can be obtained free of charge from The Cambridge Crystallographic Data Centre via www.ccdc.cam.ac.uk/ get structures. The data that support this study are available from the corresponding author upon reasonable request.

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

## Acknowledgements

This work was supported by the National Natural Science Foundation of China (22125110, 21875251, 22205233 and 21833010), the Key Research Program of Frontier Sciences of the Chinese Academy of Sciences (ZDBS-LY-SLH024), Fujian Science & Technology Innovation Laboratory for Optoelectronic Information of China (2021ZR126), the Strategic Priority Research Program of the CAS (XDB20010200), the National Postdoctoral Program for Innovative Talents (BX2021315).

## Author contributions

H.X., W.G., and Y.M. designed the experiments and prepared the single crystals; Y.L. and S.H. did the FE measurements; X.H. and L.H. fabricated the devices; X.L. performed calculations; Z.S. and J.L. conceived the idea. H.X. and Z.S. wrote this manuscript and all authors contributed to reviewing the paper.

## Competing interests

The authors declare no competing interests.
