## [Peer Review File · Nature Communications]

Record High-Tc and Large Practical Utilization Level of Electric Polarization in Metal-Free Molecular Antiferroelectric Solid SolutionsREVIEWER COMMENTS

Reviewer #1 (Remarks to the Author):

This manuscript submitted by Sun describes synthesis of the molecular-based metal-free antiferroelectric solid-solution. This work is noteworthy enough to be published by Nature Communications. Both ferroelectrics and antiferroelectrics are important phenomena in current electronics and technology, where the concept 'metal-free' by replacing metal ions by molecular ion is in line with the current global trend such as SDGs. This manuscript covers metal-free antiferroelectrics in addition to successful tuning technique of these property by composition X in halide solid solution.

I recommend this manuscript published by Nature Communications after clarifying questions from a professional background summarized in below.

1. Figure 1b is a plot of Curie temperature against a distance of DA in angstrom. There is no detailed explanation of this plot. For example, what is D and A, how Curie temperature was determined, how DA distances were defined from H-analysis. In addition, how does author determine electronegativity (upper axis) of X in solid-solutions. Is this averaged electronegativity estimated from ratio x?

2. Mechanism of structural phase transition was discussed from 'anionic displacement'. However, anion is also stated as 'stator' in the manuscript. Both meanings are opposite. Reader would be confused.

3. I do not consider that the term 'Polar ferroelectric space group' is suitable representation. Because the ferroelectric is manifested with electric switching, not originating from structure. This is also mentioned in this manuscript. Thus, 'ferroelectric space group' is not a strictly correct representation.

4. In Figure 3c, measurement points are unclear. Do all dots are data estimated experimentally using diffraction measurement? Why is there no thermal expansion of lattice?

5. I wonder Figure 5a is correct. This graph tells us that author experimentally confirm working temperature from 0 K. In this manuscript, space group was clarified only above 280 K by DSC and diffraction measurements.

6. Please show homogeneity of the proportion of x for each single crystal. Powder X-ray diffraction patterns are not enough to validate homogeneity of x, because patterns are almost independent on x.

7. Reversibility of phase transition by DSC would be desirable to be shown.

At the last, this manuscript is well-written and novel enough to satisfy a reader from Nature Communications.

Reviewer #2 (Remarks to the Author):

In the reviewed manuscript, the Authors presented the unique properties of the new compounds. The object studied is an inorganic crystal without the addition of toxic metals. It

is a simple salt-based on cyclohexylmethylammonium halides. Indeed, when most scientific groups are focused on discovering new compounds or improving the parameters of mainly organic-inorganic hybrids of transition metal halides, the results presented in this work prove that simple compounds are also exciting and worth exploring. The paper's layout is typical of communications, although the abbreviated version introduces a slight confusion, especially in part related to electrical properties. According to the experimental part, several crystals from a series of halides (Cl, Br, I) were investigated. Here we also have a description of pure and mixed crystals' properties. The first important observation is that with the I-Br-Cl series, the temperature of the phase transformation increases. This development is essential because, with the proper substitution, we can moderate the transition temperature from the active, mainly antiferroelectric phase to the paraelectric phase. Based on structural and thermal analysis, the authors constructed a phase diagram proving that the crystals undergo a two-phase transition in a certain range of $\text{Cl}_x\text{Br}_{1-x}$ concentrations. In this group of compounds, the ferroelectric phase is active in addition to the antiferroelectric phase. The double hysteresis loop measurement is noteworthy and the most outstanding achievement, which is clear evidence of the antiferroelectric phase. Nevertheless, I have some questions for the Authors:

1. Figure 4 a-c. In the figures, we have shown images of crystals obtained using a polarizing microscope. The compound studied is CMC (100%Cl). Figure (b) differs from (a). In the case of (b) we have an image of a monodomain sample, while (a) we see active domains under polarized light, which are ferroelastic ones. This effect is connected with the transition from tetragonal to orthorhombic phase, thus changing the crystallographic arrangement. We should observe two different domains in the crystal. I do not understand how the authors went from figure (b) to (c). This should be described in more detail, for example, in ESI. Because I think such information is crucial for others interested in the electrical properties of crystals. Were the electrodes attached perpendicularly to the domains? Have ferroelastic domains been observed for other crystals? And what about CMB, two transitions? What was the nature of these transitions?
2. The Authors claim that for a ratio of 0.2Br0.8Cl in Figure 4d at 408K, we observe two contributions, a ferro- and an antiferroelectric effect. In contrast, according to the phase diagram for such a composition and this temperature, we have a centrosymmetric, non-polar system. What did the P-E response look like? And what is the explanation for Figure 4d in the case of 0.8Br0.2Cl ratio?
3. I think it would be readable to create a schematic diagram that would include all the information such as transition temperatures, the nature of the transition (1st, 2nd), and the nature of the ferro-antiferro or ferroelastic-paraelastic transition. This scheme could highlight what fantastic properties these compounds have.
4. In Figure 2a (dark green), the DSC curve for CMB is a little strange. The two maxima can be observed. What does this mean?
5. I also have comments about the literature, and I think they should include references to other salts' ferroelectric properties. I would like to point out that I have not co-authored any of the following papers, but I am very familiar with them and think they fit well with the topics presented in the paper.

Piecha-Bisiorek Anna, Gaȓor Anna, Isakov Dmitry, Zieliński Piotr, Gałȓzka Mirosław, Jakubas Ryszard

Phase sequence in diisopropylammonium iodide: avoided ferroelectricity by the appearance of a reconstructed phase.

Inorganic Chemistry Frontiers, 2017, 4, 553-558

Piecha-Bisiorek Anna, Białońska Agata, Jakubas Ryszard, Zieliński Piotr, Wojciechowska

Martyna, Gałązka Mirosław

Strong improper ferroelasticity and weak canted ferroelectricity in a martensitic-like phase transition of diisobutylammonium bromide.

Advanced Materials, 2015, 27, 5023-5027

Piecha Anna, Gaḡor Anna, Jakubas Ryszard, Szklarz Przemysław

Room-temperature ferroelectricity in diisopropylammonium bromide.

CrystEngComm, 2013, 15, 940-944

In conclusion, the work is precious and will significantly interest readers looking for new materials with exciting properties. It should be reconsidered, especially the part related to electrical properties, because it will only be an excellent article conveying new knowledge about polar and non-polar properties. After the changes, I will recommend the article in Nature Communication journal.

Good luck!

Reviewer #3 (Remarks to the Author):

This paper reports the fabrication of metal-free organic perovskite mixed crystals to realize control of antiferroelectric properties. The paper discusses the correlation between the structure and the physical properties of the mixed crystals prepared by decreasing the substituent size of the organic cations from Cl and Br to I. The phase transition temperature T_c increases with decreasing substituent size from I, Br, to Cl, which is a consequence of the fact that the rotational motion of the four-fold symmetry of the organic cation is thermally less excitable. The present system is the order-disorder type 1D antiferroelectrics. The author should discuss what trends are observed when comparing the previous series of materials in terms of dimensionality and antiferroelectric energy conservation. In addition, a discussion of the correlation between substituent effects and phase transition temperatures and antiferroelectric parameters should be added from the perspective of intermolecular interactions. After the above revision, I am acceptable to publish this paper in Nature Comm.

Response to reviewers

Reviewer #1 (Remarks to the Author):

This manuscript submitted by Sun describes synthesis of the molecular-based metal-free antiferroelectric solid-solution. This work is noteworthy enough to be published by Nature Communications. Both ferroelectrics and antiferroelectrics are important phenomena in current electronics and technology, where the concept 'metal-free' by replacing metal ions by molecular ion is in line with the current global trend such as SDGs. This manuscript covers metal-free antiferroelectrics in addition to successful tuning technique of these property by composition X in halide solid solution.

I recommend this manuscript published by Nature Communications after clarifying questions from a professional background summarized in below.

Response: We deeply thank the reviewer for the time in reviewing the paper and professional feedback. The questions and suggestions are of great significance to improving our work. We have revised our manuscript according to these constructive suggestions.

Q1. Figure 1b is a plot of Curie temperature against a distance of DA in angstrom. There is no detailed explanation of this plot. For example, what is D and A, how Curie temperature was determined, how DA distances were defined from H-analysis. In addition, how does author determine electronegativity (upper axis) of X in solid-solutions. Is this averaged electronegativity estimated from ratio x ?

Response: Thanks a lot for the reviewer's comments. According to the valuable suggestion, the explanation of Figure 1b has been detailly discussed in the revision. The Curie temperatures are directly obtained from DSC and dielectric measurements (Figure 3), and the distances of $D\cdots A$ are measured from their single-crystal structures (where D is proton donor and A is acceptor in the hydrogen bonds), such as $N\cdots Cl$ distance of CMC ≈ 3.148 Å being much shorter than $N\cdots Br \approx 3.300$ Å and $N\cdots I \approx 3.529$ Å. In addition, given the mixed X^- anions with the co-occupying atomic positions, the averaged electronegativity of X^- anions in the solid solutions is estimated from the ratio x based on the known Cl/Br/I.

Please see the modified paragraph on Figure 1 Captions:

The relationship between Curie temperatures and $D\cdots A$ distance of hydrogen bonds, where donor (D) is N of organic CM^+ cation and acceptor (A) is halogen (X). The $D\cdots A$ distances can be obtained from the single-crystal structures; the averaged electronegativity of the solid solution is estimated from the ratio x based on the known Cl/Br/I.

Q2. Mechanism of structural phase transition was discussed from ‘anionic displacement’. However, anion is also stated as ‘stator’ in the manuscript. Both meanings are opposite. Reader would be confused.

Response: Great thanks for the reviewer’s valuable comments. According to the reviewer’s suggestion, we have revised the relevant paragraphs to make the expression clear in the revision. From a structural viewpoint, the stator-rotator system has been established as a promising candidate to explore antiferroelectric and/or ferroelectric compounds in the form of crystals and solid solutions. The distorted anion usually behaves as the molecular-rotator, of which the dynamic order-disordering affords a driving source to trigger phase transition. Contrarily, the halogen cation mainly stabilizes the binary structure and its anionic displacement is much weaker than the dynamic order-disordering of cations. In this sense, the anionic moiety could be regarded as the “stator-like” part. Our structure analyses also reveal that some solid solutions have the intermediate phase determined as a polar space group of $P4mm$, and the source of antiferroelectric-ferroelectric-paraelectric phase transition mainly originates from the dynamic ordering and antipolar reorientation of organic cationic rotators.

The modified paragraph on Page 2:

The family of $CMBr_xI_{1-x}$ and $CMBr_xCl_{1-x}$ solid solutions adopt a hydrogen-bonding dimer motif, resembling the binary stator-rotator system with AFE-to-PE phase transition (Fig. 1a). Organic CM^+ cation is the dynamically active moiety with order-disorder genius like “rotator”, while halogen anions linked to cations via $N-H\cdots X$ hydrogen bonds stabilize the structure. The cooperativity of dipolar components in the adjacent crystal lattices afford the power source to create long-range AFE orders, mainly determined by the cationic reorientation.

The modified paragraph on Page 3:

“.... The halogen anions of these solid solutions exhibit relative atomic displacements in FEP, thus coinciding with the variation of crystal symmetry.²⁴ Hence, the origin of AFE-FE-PE phase transitions can ascribe to the collaboration between organic cationic ordering and anionic displacement.

Q3. I do not consider that the term ‘Polar ferroelectric space group’ is suitable representation. Because the ferroelectric is manifested with electric switching, not originating from structure. This is also mentioned in this manuscript. Thus, ‘ferroelectric space group’ is not a strictly correct representation.

Response: Thanks for the reviewer’s professional suggestion. As the reviewer mentioned, the term “ferroelectric space group” has been modified in the revised manuscript.

The modified paragraph on Page 3:

“..., of which the intermediate phase is determined as a polar space group of $P4mm$.”

Q4. In Figure 3c, measurement points are unclear. Do all dots are data estimated experimentally using diffraction measurement? Why is there no thermal expansion of lattice?

Response: Thanks for the reviewer’s comments. The description in Figure 3 is not very unambiguous, and the relevant caption has been modified in the revised manuscript. Figure 3c shows the diagram of temperature dependence *versus* lattice parameters, which are obtained directly from the single-crystal diffraction of cell parameters at different temperatures (as depicted by the short dash lines). Moreover, thermal expansion of cell lattices is also observed within the single phases, including either antiferroelectric phase or paraelectric phase. For instance, the volume of CMB expands from 1948.2 (at 298 K) to 1989.0 (at 360 K) in the antiferroelectric phase, and then drastically shrink to 501.7 (380 K) in the paraelectric state caused by phase transition. It is the drastic volume change during the antiferroelectric-to-paraelectric phase transition that makes the thermal expansion inconspicuous.

The modified on **Figure 3c**:

Fig. 3 / c, Schematic diagram of temperature dependence versus cell lattice parameters obtained directly from the X-ray single-crystal diffraction, as depicted by the short dash lines.

Q5. I wonder Figure 5a is correct. This graph tells us that author experimentally confirm working temperature from 0 K. In this manuscript, space group was clarified only above 280 K by DSC and diffraction measurements.

Response: Thanks for the reviewer’s useful suggestion. As the reviewer mentioned, the space

group of our antiferroelectric is clarified above 280 K by DSC and diffraction measurements. For the clarity and accuracy of Figure 5a, we have modified this paragraph in the revision. The Curie temperature (T_c) values for CMC and some reported molecule-based AFE compounds have been labelled in the revised Figure 5. The columnar with twill background present the temperature range for experimentally measuring the antiferroelectric properties, such as the characteristic double P - E hysteresis loop.

The modified on **Figure 5a**:

Fig. 5 / a, AFEP Curie temperature (T_c) of CMC compared with some reported molecule-based AFE compounds. The columnar with twill background present the temperature range for experimentally measuring the antiferroelectric properties.

Q6. Please show homogeneity of the proportion of x for each single crystal. Powder X-ray diffraction patterns are not enough to validate homogeneity of x , because patterns are almost independent on x .

Response: Thanks for the reviewer's useful suggestion. As the reviewer mentioned, the characterizing the homogeneity of the proportion of x for solid-solution single crystals is very important for studying the properties of the materials. Generally, EDS elemental mapping is a favorable method to characterize the homogeneity of materials. As shown in Supplementary Figure 10-12, the distribution of C, N, Cl, Br, and I in the whole selected area reveals that the elements are uniformly dispersed on the single crystals.

Supplementary Fig. 10 | FESEM images and EDS elemental mapping of $\text{CMBr}_{0.8}\text{Cl}_{0.2}$.

Supplementary Fig. 11 | FESEM images and EDS elemental mapping of $\text{CMBr}_{0.2}\text{Cl}_{0.8}$.

Supplementary Fig. 12 | FESEM images and EDS elemental mapping of $\text{CMBr}_{0.5}\text{I}_{0.5}$.

Q7. Reversibility of phase transition by DSC would be desirable to be shown.

Response: Thanks for the reviewer's useful suggestion. The reversibility of phase transition by DSC can be seen in Supplementary Figure 6.

Supplementary Fig. 6 | DSC curves with the heating/cooling rate of 20 K/min for the selected compositions.

At the last, this manuscript is well-written and novel enough to satisfy a reader from Nature Communications.

=====**The end of reply to Reviewer 1**=====

Reviewer #2 (Remarks to the Author):

In the reviewed manuscript, the Authors presented the unique properties of the new compounds. The object studied is an inorganic crystal without the addition of toxic metals. It is a simple salt-based on cyclohexylmethylammonium halides. Indeed, when most scientific groups are focused on discovering new compounds or improving the parameters of mainly organic-inorganic hybrids of transition metal halides, the results presented in this work prove that simple compounds are also exciting and worth exploring. The paper's layout is typical of communications, although the abbreviated version introduces a slight confusion, especially in part related to electrical properties. According to the experimental part, several crystals from a series of halides (Cl, Br, I) were investigated. Here we also have a description of pure and mixed crystals' properties. The first important observation is that with the I-Br-Cl series, the temperature of the phase transformation increases. This development is essential because,

with the proper substitution, we can moderate the transition temperature from the active, mainly antiferroelectric phase to the paraelectric phase. Based on structural and thermal analysis, the authors constructed a phase diagram proving that the crystals undergo a two-phase transition in a certain range of $\text{Cl}_x\text{Br}_{1-x}$ concentrations. In this group of compounds, the ferroelectric phase is active in addition to the antiferroelectric phase. The double hysteresis loop measurement is noteworthy and the most outstanding achievement, which is clear evidence of the antiferroelectric phase. Nevertheless, I have some questions for the Authors:

Response: We deeply thank the reviewer for the time in reviewing the paper and professional feedback. The questions and suggestions are very important to improve our work. We have carefully revised our manuscript according to these constructive suggestions.

Q1. Figure 4 a-c. In the figures, we have shown images of crystals obtained using a polarizing microscope. The compound studied is CMC (100%Cl). Figure (b) differs from (a). In the case of (b), we have an image of a monodomain sample, while (a) we see active domains under polarized light, which are ferroelastic ones. This effect is connected with the transition from tetragonal to orthorhombic phase, thus changing the crystallographic arrangement. We should observe two different domains in the crystal. I do not understand how the authors went from figure (b) to (c). This should be described in more detail, for example, in ESI. Because I think such information is crucial for others interested in the electrical properties of crystals. Were the electrodes attached perpendicularly to the domains? Have ferroelastic domains been observed for other crystals? And what about CMB, two transitions? What was the nature of these transitions?

Response: We deeply thank the reviewer for the professional reviewing. According to the valuable suggestion, more details of Figure 4 a-c have been described in the Supplementary Information. The conoscopic images and ferroelastic domains were measured on Nikon Eclipse LV 100N POL. Figures 4 (b) and (c) show the preparation of electrodes according to the relationship between optical axis and the direction of electric polarization. In detail, we can determine the optical axial plane by observing the dual optical axes of biaxial crystal under the conoscope. Based on the electric polarization direction of the orthorhombic system perpendicular to the optical axial plane, we can preliminarily determine the polarization direction and prepare the electrodes. The arrowheads denote the polarization axis rather than the direction of the domain.

In addition, ferroelastic domains of this binary solid-solution family of $\text{CMBr}_x\text{I}_{1-x}$ and $\text{CMBr}_x\text{Cl}_{1-x}$ have been provided in Supplementary Figure 9. For CMB, it displays the occurrence of AFE-FE-PE successive phase transitions. (Supplementary Figure 7). The mechanisms of two-step phase transitions are determined as antiferroelectric-ferroelectric-paraelectric transitions, respectively (Supplementary Figure S7; Ref. 24). For the first

antiferroelectric-to-ferroelectric phase transition, the methylamino group attached to the six-member carbocyclic ring is prone to the dynamic order-disorder change that affords the driving force to phase transition. This phase transition is classified as an order-disorder type. For the second ferroelectric-to-paraelectric one, its crystal structure becomes completely disordered and thus possesses the center inversion symmetry, which leads to the disappearance of bulk electric polarization ($P_s = 0$). It is the partial frozen ordering and relative atomic displacement that account for the second phase transition.

Supplementary Fig. 9 | Ferroelastic domains observed in the binary solid-solution family of $\text{CMBr}_x\text{I}_{1-x}$ and $\text{CMBr}_x\text{Cl}_{1-x}$.

Q2. The Authors claim that for a ratio of 0.2Br0.8Cl in Figure 4d at 408K, we observe two contributions, a ferro- and an antiferroelectric effect. In contrast, according to the phase diagram for such a composition and this temperature, we have a centrosymmetric, non-polar system. What did the P - E response look like? And what is the explanation for Figure 4d in the case of 0.8Br0.2Cl ratio?

Response: Thanks a lot for the reviewer's comments. As the reviewer referred, an additional current peak emerges in $\text{CMBr}_{0.2}\text{Cl}_{0.8}$, suggesting the coexistence of both AFE and FE orders that resemble some other inorganic solid solutions (Please see *Acta Mater.* **2019**, 179, 255; *J. Mater. Chem. A*, **2020**, 8, 2369). This observation might be explained by an electric-field-induced AFE-FE phase transition. The relevant references have been cited in the revised manuscript. According to the reviewer's suggestion, we have also measured the P - E loops, which suggest the coexistence of both AFE and FE orders in $\text{CMBr}_{0.2}\text{Cl}_{0.8}$. (Please see Figure S1 in this letter)

Figure S1. Ferroelectric hysteresis loop and current *versus* electric field traces of $\text{CMBr}_{0.2}\text{Cl}_{0.8}$.

According to the reviewer's suggestion, we have measured the P - E loops. A pair of obvious ferroelectric characteristic current peaks of $\text{CMBr}_{0.8}\text{Cl}_{0.2}$ solidly confirm the ferroelectricity in the intermediate phase $P4mm$. (Please see Figure S2 in this letter)

Figure S2. Ferroelectric hysteresis loop and current *versus* electric field traces of $\text{CMBr}_{0.8}\text{Cl}_{0.2}$.

Q3. I think it would be readable to create a schematic diagram that would include all the information such as transition temperatures, the nature of the transition (1st, 2nd), and the nature of the ferro-antiferro or ferroelastic-paraelastic transition. This scheme could highlight what fantastic properties these compounds have.

Response: Thanks for the reviewer's useful suggestion. We agree with the reviewer's statement that it makes sense to create a schematic diagram containing the phase transition information. Therefore, the phase transition properties of the solid solutions have been plotted in the schematic (Supplementary Figure 7).

Supplementary Fig. 7 | Phase transformation and space groups of $\text{CMBr}_x\text{I}_{1-x}$ and $\text{CMBr}_x\text{Cl}_{1-x}$ solid solutions.

Q4. In Figure 3a (dark green), the DSC curve for CMB is a little strange. The two maxima can be observed. What does this mean?

Response: Thanks a lot for the reviewer's comments. The DSC curve for CMB exhibit two pairs of reversible exothermic/ endothermic peaks at 364/355 K (T_1) and 368/365 K (T_2) in the heating and cooling model, displaying the occurrence of AFE-FE-PE successive phase transitions. The mechanisms of two-step phase transitions are determined as antiferroelectric-ferroelectric-paraelectric transitions, respectively (Supplementary Figure S7; Ref. 24). For the first antiferroelectric-to-ferroelectric phase transition, the methylamino group attached to the six-member carbocyclic ring is prone to the dynamic order-disorder change that affords the driving force to phase transition. This phase transition is classified as order-disorder type. For the second ferroelectric-to-paraelectric one, its crystal structure becomes completely disordered and thus possesses the center inversion symmetry, which leads to the disappearance of bulk electric polarization ($P_s = 0$). It is the partial frozen ordering and relative atomic displacement that account for the second phase transition.

Q5. I also have comments about the literature, and I think they should include references to other salts' ferroelectric properties. I would like to point out that I have not co-authored any of

the following papers, but I am very familiar with them and think they fit well with the topics presented in the paper.

Piecha-Bisiorek Anna, Gaḡor Anna, Isakov Dmitry, Zieliński Piotr, Gałazka Mirosław, Jakubas Ryszard

Phase sequence in diisopropylammonium iodide: avoided ferroelectricity by the appearance of a reconstructed phase.

Inorganic Chemistry Frontiers, 2017, 4, 553-558

Piecha-Bisiorek Anna, Białońska Agata, Jakubas Ryszard, Zieliński Piotr, Wojciechowska Martyna, Gałazka Mirosław

Strong improper ferroelasticity and weak canted ferroelectricity in a martensitic-like phase transition of diisobutylammonium bromide.

Advanced Materials, 2015, 27, 5023-5027

Piecha Anna, Gaḡor Anna, Jakubas Ryszard, Szklarz Przemysław

Room-temperature ferroelectricity in diisopropylammonium bromide.

CrystEngComm, 2013, 15, 940-944

Response: Thanks for the reviewer's suggestion. We have carefully read the recommended papers that closely relate to our topics. The recommended references have been cited in the appropriate positions in the revised version.

In conclusion, the work is precious and will significantly interest readers looking for new materials with exciting properties. It should be reconsidered, especially the part related to electrical properties, because it will only be an excellent article conveying new knowledge about polar and non-polar properties. After the changes, I will recommend the article in Nature Communication journal.

===== **The end of reply to Reviewer 2** =====

Reviewer #33 (Remarks to the Author):

This paper reports the fabrication of metal-free organic perovskite mixed crystals to realize control of antiferroelectric properties. The paper discusses the correlation between the structure and the physical properties of the mixed crystals prepared by decreasing the substituent size of the organic cations from Cl and Br to I. The phase transition temperature T_c increases with decreasing substituent size from I, Br, to Cl, which is a consequence of the fact that the rotational motion of the four-fold symmetry of the organic cation is thermally less excitable. The present system is the order-disorder type 1D antiferroelectrics. The author should discuss what trends are observed when comparing the previous series of materials in

terms of dimensionality and antiferroelectric energy conservation. In addition, a discussion of the correlation between substituent effects and phase transition temperatures and antiferroelectric parameters should be added from the perspective of intermolecular interactions. After the above revision, I am acceptable to publish this paper in Nature Comm.

Response: We deeply thank the reviewer for the time in reviewing the paper and professional feedback. The questions and suggestions are very important to improve our work. We have carefully revised our manuscript according to these constructive suggestions.

1. The author should discuss what trends are observed when comparing the previous series of materials in terms of dimensionality and antiferroelectric energy conservation.

Response: Thanks a lot for the reviewer's comments. In terms of antiferroelectric energy conservation, the breakdown electric field, the switching field (E_A and E_F), electric polarization and thermal stability of materials are more important than the dimensions of the compound. (Please see *Adv. Funct. Mater.* **2018**, 1803665; *Prog. Mater. Sci.*, **2014**, 63, 1-57) The comparison of the energy conservation properties among different antiferroelectric materials has been summarized in Supplementary Table 10.

Supplementary Table 10 | Comparison of the energy conservation properties among different antiferroelectric materials.

Material	Form	E (kV/cm)	W_s (J/cm ³)	W_{te} (J/cm ³)	η (%)	Reference
This work	single crystal	63	0.54	0.28	52	This work
CMB	single crystal	57	0.23	0.1	43	This work
[H-55dmbp][Hca] salt	single crystal	173	0.56	0.51	90	1
(i-BA) ₂ CsPb ₂ Br ₇	single crystal	94	0.38	0.26	63	7
(BA) ₂ (EA) ₂ Pb ₃ I ₁₀	single crystal	60	0.18	0.15	83	8
AgNbO ₃	ceramic	184	7	2.8	40	11
Pb _{0.97} La _{0.02} (Zr _{0.50} Sn _{0.39} Ti _{0.11})O ₃	ceramic	130	3	1.5	50	23
0.95NBT-0.05BH	ceramic	125	2.7	1.3	48.2	24
Hf _{0.3} Zr _{0.7} O ₂	film	3260	55	28	51	25

2. In addition, a discussion of the correlation between substituent effects and phase transition temperatures and antiferroelectric parameters should be added from the perspective of intermolecular interactions.

Response: Great thanks for the reviewer's comment. According to this useful suggestion, the correlation between substituent effects and phase transition temperatures and antiferroelectric parameters has been detailly discussed in the revision (Fig. 1b). Structurally, the N-H...X hydrogen bonds are the main intermolecular interactions in the binary solid-solution family of CMBr_xI_{1-x} and CMBr_xCl_{1-x}. Thus, modulating ion radius and electronegativity of the halogen anions could significantly change the N-H...X distances and further change the strength of H-bonds. For example, The N...Cl distance of CMC \approx 3.148 Å is much shorter than N...Br \approx

3.300 Å and $N\cdots I \approx 3.529$ Å. The tighter the stator bound to the organic CM^+ cation (rotor) by hydrogen bonds, the harder it will be for the rotator to rotate, which indicates the dynamic ordering and reorientation of C-N bonds and six-member rings of two adjacent bipartite arrays require much more energy to overcome barriers. Consequently, A molecule-level I/Br/Cl halogen modification has effectively enhanced the Curie temperature and antiferroelectric properties, as demonstrated by Figure 1b in the revision.

=====**The end of reply to Reviewer 3**=====

REVIEWERS' COMMENTS

Reviewer #2 (Remarks to the Author):

The Authors responded to my suggestions comprehensively. Also, this applies to the responses to the requests of other Reviewers. I found the work to be of great value. I enjoyed reading it, and seeing the hysteresis loop where we have two FE-AFE effects is super!

Thank You, and good luck with your future work.

Reviewer #3 (Remarks to the Author):

I think that the paper was improved after the revisions. The author carefully responses for the comments from three reviewers. I recomend to be published in Nature Commum.

Response to reviewers

Reviewer #2 (Remarks to the Author):

The Authors responded to my suggestions comprehensively. Also, this applies to the responses to the requests of other Reviewers. I found the work to be of great value. I enjoyed reading it, and seeing the hysteresis loop where we have two FE-AFE effects is super!

Thank You, and good luck with your future work.

Response: We thank the Reviewer for their positive assessment of our work.

Reviewer #3 (Remarks to the Author):

I think that the paper was improved after the revisions. The author carefully responses for the comments from three reviewers. I recomend to be published in Nature Commum.

Response: We thank the Reviewer for their positive assessment of our work.